# NEXUS: SPECIALIZATION MEETS ADAPTABILITY FOR EFFICIENTLY TRAINING MIXTURE OF EXPERTS

## ABSTRACT

Efficiency, specialization, and adaptability to new data distributions are qualities that are hard to combine in current Large Language Models. The Mixture of Experts (MoE) architecture has been the focus of significant research because its inherent conditional computation enables such desirable properties. In this work, we focus on "upcycling" dense expert models into an MoE, aiming to improve specialization while also adding the ability to adapt to new tasks easily. We introduce Nexus, an enhanced MoE architecture with *adaptive routing* where the model learns to project expert embeddings from domain representations. This approach allows Nexus to flexibly add new experts after the initial upcycling through separately trained dense models, without requiring large-scale MoE training for unseen data domains. Our experiments show that Nexus achieves a relative gain of up to 2.1% over the baseline for initial upcycling, and a 18.8% relative gain for extending the MoE with a new expert by using limited finetuning data. This flexibility of Nexus is crucial to enable an open-source ecosystem where every user continuously assembles their own MoE-mix according to their needs.

## 1 INTRODUCTION

In an era of bigger and bigger models (Canziani et al., 2016; Strubell et al., 2019; Rae et al., 2021; Raffel et al., 2020; Bommasani et al., 2022; Hooker, 2024), there are several key objectives driving state-of-art progress. Doing *more with less* by improving efficiency (Treviso et al., 2023) remains paramount, but in addition to efficiency, the deployment of these models in the wild means that the ability to adapt to new data (Pozzobon et al., 2023b; Gururangan et al., 2020a; Jang et al., 2022; Jin et al., 2022), and specialization of compute (Zadouri et al., 2024; Shazeer et al., 2018; Riquelme et al., 2021; Du et al., 2022; Fedus et al., 2022) have gained renewed focus. While all these properties are desirable, a formidable challenge is designing architectures that can fulfill *all* of these requirements.

The Mixture of Experts (MoE) approach gained prominence because of its efficiency properties. In contrast to dense models which require significant compute to deploy, MoE approaches only activate a subset of the parameters for every single token. Intuitively, not all parameters are necessary for each request, as some parameters will specialize on certain tasks, and those unrelated to the current request can be ignored. However, while MoEs greatly improved efficiency, the ability to induce meaningful specialization has been more limited, with observations that experts don't appear to exhibit dedicated expertise (Jiang et al., 2024; Zoph et al., 2022; Zadouri et al., 2023). Furthermore, MoEs tend to suffer from severe training instabilities (Zoph et al., 2022).

Recent work has attempted to address both the training instabilities and the lack of specialization. These techniques often train completely separate experts and "upcycle" (combine) them into a single unified MoE model *after* dense training (Sukhbaatar et al., 2024). This reduces the memory and communication cost, and improves ***efficiency*** during training as computations are more local and cross-device communication is reduced (Li et al., 2022; Gururangan et al., 2023). Notably, the other major advantage of these approaches is the increase in ***specialization*** with separate experts that are trained on specific domains, making them clearly responsible for their human-interpretable subset of the data. On the other hand, MoEs with a standard router, which needs to be trained on a mix of all training data, are not designed to maintain domain specialization (Jiang et al., 2024).

However, efficiently integrating new experts into upcycled MoE models - a setting that is of great interest for ***adaptability*** objectives - is far less studied. For most practitioners, given the scale of

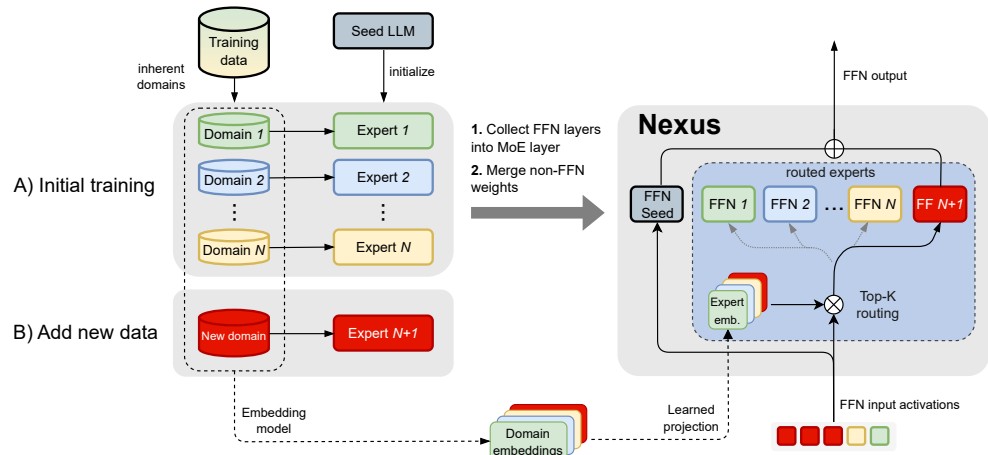

Figure 1: **Depiction of Nexus for a single Transformer block: A)** In the initial training phase, each expert is trained separately. Its training data is embedded by an embedding model and stored. The experts are combined by initializing each block's MoE layer with the expert FFNs, and finetuning the model on a mix of all domains. During a forward pass, the seed model FFN is used as shared expert and always activated. For the other experts, we perform top-1 routing based on the similarity of the input data with the transformed expert embeddings, which is equivalent to viewing the learned projection as a hypernetwork whose output is the router weight matrix. **B)** Later, we can add a new expert by appending its training data embedding to the existing domain embeddings. The router function is independent of the number of experts, and therefore adapts fast to the new one.

modern LLMs (Brown et al., 2020; Touvron et al., 2023; Kaplan et al., 2020; Anil et al., 2023) training MoEs repeatedly is an infeasible computational cost. Furthermore, most model development fails to take into account distribution drift in use cases, with limited flexibility and applicability across different tasks and domains (Pozzobon et al., 2023a; Gururangan et al., 2020b). However, human language is shaped by a cumulative culture, constantly building upon itself and evolving over time (Silvey, 2016). Also, specialized use cases such as multilingual, code and math often require tailored additional training.

In this work, we attempt to reconcile all three desirable properties: *efficiency, specialization, and adaptability.* We ask *"how can we adaptively combine separately trained specialized experts?"* To address this, we introduce **Nexus**, a novel MoE architecture that parameterizes the router based on domain-specific data by learning to project the embedding of each data domain to an expert embedding. This learnable projection for the router allows for the easy extension of the MoE model with new experts that are trained independently on new datasets of interest. This also avoids the difficulties of MoE training, as our learned router scales with the number of experts without needing to be trained from scratch, which enables adding or removing experts as desired.

Our experiments show that Nexus outperforms previous work when upscaling an MoE from separately trained specialized domain experts. Going beyond the single upscaling phase, Nexus can be efficiently extended with a new expert trained on a new domain, by finetuning it with much fewer tokens, compared to the finetuning after the initial upcycling.

In summary, our contributions are as follows:

1. We present Nexus, a novel MoE framework designed to enhance sparse upcycling of specialized dense experts, while reducing the training cost of MoEs by facilitating easy adaptation to unseen data distributions. In Nexus, the traditional linear router from vanilla MoE models is replaced with routing based on the similarity of layer inputs to an expert embedding vector, derived from the average embedding of the corresponding expert dataset.

2. Our method outperforms the existing approach for upcycling specialized models into MoE, leading to 2.1% and 1.6% relative increase over the upcycled MoE (linear router) in 470M and 2.8B scales respectively. This enables performance increase in general tasks with 5.8% and 7.4% relative gains over the dense seed model at 470M and 2.8B respectively.

3. Our method enables efficient adaptation to new domains by extending upcycled MoE with the new experts trained on unseen datasets. In this setting, Nexus outperforms the baseline MoE (linear router) when finetuning on the limited amount of data, leading 18.8% relative gain on the new domain with 1B finetuning tokens upon MoE extension.

4. Finally, we show that our method is robust across different load balancing and data mixtures, and consistently outperforms the MoE with a linear router for specialized upcycling, confirming the benefits of the *adaptive* routing based on domain projections used in Nexus.

## 2 BACKGROUND

Sparse Mixture of Experts architectures (Shazeer et al., 2017; Fedus et al., 2022) replace the feed-forward network (FFN) with an MoE layer in the Transformer block (Vaswani et al., 2017). An MoE layer consists of a router network $R$ and a set of $n$ experts, $E_1, ..., E_n$, where each expert $E_i$ corresponds to an independent dense feed-forward network. The router network $R$ is commonly parameterized by trainable weights $W_r \in \mathbb{R}^{h \times n}$ where $h$ is the model hidden dimension, and followed by a *softmax* function which takes an intermediate token representation $x$ as input and combines the output of each expert based on the gating scores $s_1, ..., s_n$. Sparse MoEs only use the *top-k* experts $E_k$ based on experts gating scores $s_i$.

$$s_i = R(x) = \text{softmax}(W_r^T x) \qquad \text{(Router)}$$
$$s_k = \text{TopK}(s_i) \qquad \text{(Top-K Routing)}$$
$$y = \sum_{i=1}^{k} s_k \cdot E_k(x) \qquad \text{(MoE)}$$

Sparse Upcycling (Komatsuzaki et al., 2023) initializes an MoE model from a dense Transformer model by copying FFN layers as MoE experts, and the router layer is trained from scratch. BTX (Sukhbaatar et al., 2024) generalize this approach to initialize each MoE expert from the FFN layer of a different dense model, and all other parameters are averaged over the dense models.

In **Nexus**, we leverage upcycling specialized expert models similar to BTX, however, it diverges in terms of MoE training, in particular with its novel MoE router, which enables to efficiently extend the MoE in multiple rounds after the sparse upcycling. We describe our method in the next section.

## 3 ADAPTIVE ROUTER FOR UPCYCLING SPECIALIZED EXPERTS AS MOE

The core component of an MoE model is the router, as it determines which experts to activate for any given input. In vanilla MoEs, the router is a learned linear layer that takes the token intermediate representations as input and computes the expert probabilities. However, this router does not necessarily learn specialization as MoEs are commonly trained using an auxiliary load balancing loss to improve training stability (Fedus et al., 2022; Jiang et al., 2024). In Nexus, we propose a novel MoE router where per MoE block we learn a projection layer from given pre-computed domain embeddings to expert embeddings. We parametrize this projection layer $P_r$ as a two-layer MLP with a SwiGLU activation function (Shazeer, 2020):

$$e_i = P_r(d_i) \qquad \text{(Domain to Expert Embeddings)}$$
$$= W_2 \cdot \text{SwiGLU}(W_1 \cdot d_i)$$

where $d_i \in \mathbb{R}^m$, and $e_i \in \mathbb{R}^h$ are the domain and expert embeddings for the $i$th domain respectively., where $m$ and $h$ are the domain embedding and the model dimensions. $W_1 \in \mathbb{R}^{2h \times d}, W_2 \in \mathbb{R}^{l \times l}$ are linear layers, and SwiGLU is defined as $\mathbb{R}^{2n} \to \mathbb{R}^n$. Given the expert embeddings $e_i$ and layer inputs $x \in \mathbb{R}^{s \times h}$, we then compute routing probabilities $s_i$ as:

$$s_i = \text{softmax}(x \cdot e_i) \qquad \text{(Routing Scores)}$$

Unlike the standard router, Nexus's router includes a stronger inductive bias through pre-computed domain embeddings[1] that enables expert embedding to specialize. Thus, $x \cdot e_i$ gives a high value for input tokens that are closer to the domain of the corresponding expert. Notably, this router is particularly suited for the sparse upcycling setting where the dense experts are separately trained on different domains.

**Connection to hypernetworks.** Our router parametrization is closely related to hypernetworks (Ha et al., 2016) as the projection layer $P_r$ generates parameters for the router during runtime for a given input. We use domain embeddings as the input to the projection layer, enabling efficient adaptation and also a better cross-domain transfer based on the similarity between domain embeddings as shown in previous work (Mahabadi et al., 2021; Üstün et al., 2022).

**Upcycling dense experts as an MoE.** After training dense expert models, we merge the individual experts into a unified MoE by appending their FFNs along a new dimension to create an MoE layer per Transformer block. Unlike Sukhbaatar et al. (2024), instead of using the original FFN of the seed model as one of the routed experts in an MoE layer, we use it as the "shared expert" $\text{FFN}_s$ (Rajbhandari et al., 2022; Dai et al., 2024) to better preserve the previous capabilities in the MoE model. For all non-FFN parameters including the attention weights, we merge expert parameters using simple weight averaging:

$$\text{FFN}_{moe} = \text{FFN}_s + [\text{FFN}e_1, \text{FFN}e_2, ..., \text{FFN}e_n] \qquad \text{(MoE Layer FFNs)}$$

$$\phi_{moe} = \frac{\sum_{i=1}^{n} \phi_i}{n} \qquad \text{(Merge Non-FFN params.)}$$

**Efficient adaptation to new domains.** An important advantage of method is that when a new data domain is present after MoE training, we use the learned projection $P_r$ to compute expert embedding of the new domain as $e_{new} = P_r(d_{new})$. This enables to enhance the trained MoE model with additional dense experts, which are trained in the same way as the initial experts. The FFN parameters of the new expert are simply appended to the array of existing experts.

To adequately preserve the non-FFN parameters of existing experts, we perform a weighted average $\phi_f = (1 - \lambda) \cdot \phi_{moe} + \lambda \cdot \phi_{new}$ where $\phi_f$, $\phi_e$, and $\phi_{moe}$ are parameters of the final MoE, dense expert, and initial MoE model and $\lambda = 1/(n + 1)$. This enables *efficiently adapting* Nexus to new domain by extending it with the new dense expert trained independently. After extending the MoE with a new expert, we perform a lightweight finetuning with a limited number of tokens.

## 4 EXPERIMENTS

### 4.1 EXPERIMENTAL SETTING

Our experimental setup includes 3 phases. Figure 1 shows the architecture of Nexus and the corresponding experimental setting:

**1. Training specialized expert LMs.** For training the dense specialized experts, we use the sub-datasets from the SlimPajama dataset (Soboleva et al., 2023), a 627B token English-language corpus assembled from web data of various sources. We initialize four dense experts from the weights of the seed model and train them on the ARXIV, BOOKS, C4, GITHUB, STACKEXCHANGE, and WIKIPEDIA domains.[2] As the seed model, we use 470M and 2.8B parameters decoder-only autoregressive Transformer models (Radford et al., 2019), each of them trained with a standard language modeling objective for 750B tokens. We train dense experts for 20 and 40 billion tokens for 470M and 2.8B seed models respectively. We use parallel attention layers, (Anil et al., 2023; Wang, 2021), SwiGLU activation (Shazeer, 2020), no biases in dense layers, and a byte-pair-encoding (BPE) tokenizer with a vocabulary size of 256,000. During training, we use a linear warmup (10% of total steps) to a maximum learning rate of 1e-3 and a cosine decay schedule to 3e-4.

---

[1]We used Cohere Embed v3 (Cohere, 2023) as an external embedding model to compute domain embeddings based on individual data sources. However, similar to Gururangan et al. (2023), pre-training data can also be clustered and the centroids can be used for domain embeddings.

[2]We exclude the Github and StackExchange datasets from SlimPajama in order to ablate adding a new expert model using the CODE domain

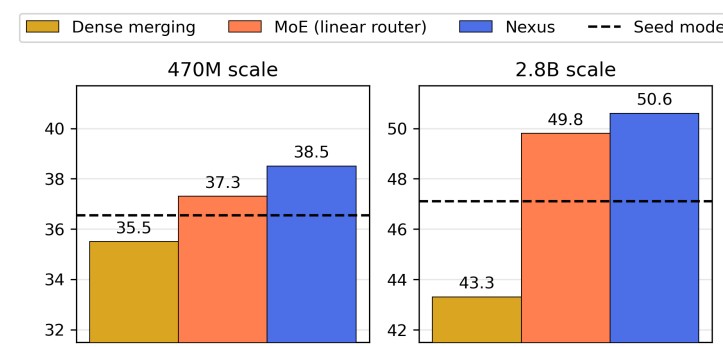

Figure 2: **Downstream performance at different scales:** `Nexus` consistently outperforms upcycled baselines on both the 470M and 2.8B parameters scale, showing the robustness of our method. We report the average performance on Knowledge, Science, Reasoning and MMLU.

**2. MoE training.** After the training of dense expert models, we merge them into a unified MoE by appending their FFNs along a new dimension to create an MoE layer per Transformer block. For the *shared expert* in our MoE layer, we use the original FFN layer of the seed model to better preserve the previous capabilities in the MoE model. For all non-FFN parameters including the attention weights, we merge expert parameters using simple weight averaging, following Sukhbaatar et al. (2024). After the MoE model is created, we continually train it for an additional 25B and 40B tokens respectively for the 470M and 2.8B experiments, on a mix of all domain and original pre-training datasets, using the same training hyperparameters as in the single expert training. Finally, we train the MoE models using an additional 1B tokens by upweighting the original pre-training dataset as it includes high-quality data sources such as instruction-style datasets using a cosine learning rate decay to 3e-5 (Parmar et al., 2024).

**3. Extending the MoE model with new experts.** After adding a new expert as defined in Section 3, we finetune the extended MoE model for up to 1 billion tokens using a uniformly sampled data mix consisting of 50% the previous domains and pre-training data and 50% the new domain. For the new expert (CODE), we train a dense model using code documents from StarCoder (Li et al., 2023) with the same settings as for the training of the initial experts. As the 470M scale MoE did not have sufficient instruction following capabilities to attempt the code benchmarks, we only tested extending the MoEs with a new expert on the 2.8B scale.

## 4.2 BASELINES

We compare our experiments against two baselines:

**Dense Merging** where all separately pre-trained experts and the seed model merged into a dense Transformer via equal weight averaging similar to BTM (Li et al., 2022). This allows us to ask *What are the benefits of routing MoE over simple averaging?*

**MoE (Linear Router)** which is an MoE with a standard linear router that is upcycled from dense experts, to evaluate Nexus's novel router for upcycling. Here, we ask *how does our specialized routing compare to conventional learned linear routing?* For a fair comparison, we also train this MoE model on the same datasets and for the same number of tokens as our method, and use the same architectural modifications such as shared experts.

## 4.3 EVALUATION

For the downstream evaluation, we measure the performance of each model on 15 tasks from five evaluation categories that reflect different capabilities based on the tasks and the datasets used in the benchmarks.

These task categories are **(1) Knowledge**, to measure question-answering capabilities based on world knowledge and web documents such as Wikipedia, we report the performance on OpenBookQA (Mihaylov et al., 2018), Natural Questions (Kwiatkowski et al., 2019), TriviaQA (Joshi et al., 2017),

|  | **Know.** | **Science** | **Reason.** | **MMLU** | **Code** (excl. in upcyc.) | **Avg.** (w/o Code) |
|---|---|---|---|---|---|---|
| SEED MODEL (2.8B) | 27.1 | 62.0 | **63.8** | 35.4 | **8.4** | 47.1 |
| **Upcycled Models** | | | | | | |
| DENSE MERGING | 17.6 | 60.3 | 59.2 | 36.0 | 3.4 | 43.3 |
| MoE (LINEAR ROUTER) | 31.5 | 66.5 | 62.9 | 38.6 | 2.6 | 49.8 |
| NEXUS | **33.2** | **67.3** | 62.6 | **39.4** | 2.7 | **50.6** |

Table 1: **Downstream task results for Nexus with a 2.8B parameter seed model.** Our approach outperforms the baselines in 3 out of 4 evaluation categories. `Dense merging` corresponds a dense model with 2.8B parameters, while both `Nexus` and `MoE (linear router)` have 4.3B active and 9.1B total parameters. Note that the trained models show severe forgetting on code benchmarks, as we exclude CODE data on purpose during the upcycling phase to simulate extending models with a new dataset in Section 5.2.

QUAC (Choi et al., 2018) (all 0-shot) and SQuAD (4-shot) (Rajpurkar et al., 2016). **(2) Science**, to measure knowledge in science-oriented academic benchmarks, we use ARC-Easy, ARC-Challenge (Clark et al., 2018), SciQ (Welbl et al., 2017) (all 0-shot). **(3) Reasoning**, we use CommonSenseQA (Talmor et al., 2019), SIQA (Sap et al., 2019), PIQA (Bisk et al., 2020), WinoGrande (Sakaguchi et al., 2019), and HellaSwag (Zellers et al., 2019) (all 0-shot). **(4) General Language Understanding**, we use MMLU (5-shot) (Hendrycks et al., 2021). **(5) Code**, we evaluate models on MBPP (Austin et al., 2021), LBPP (Matton et al., 2024) and HumanEval-Pack (Chen et al., 2021) that includes Cpp, Javascript, Java, Go, Python, and Rust (all 0-shot).

## 5 RESULTS AND DISCUSSION

### 5.1 MAIN RESULTS FOR UPCYCLED MODELS

We first compare `Nexus` to the upcycled baselines `MoE with linear router` and `dense merging`. Here, we ask "*How does our MoE upcycling recipe with adaptive routing compare against baseline upcycling approaches?*"

**470M parameter seed model.** Table 4 (Appendix D) shows performances of upcycled models including `Nexus` where a 470M seed model is used to train dense experts. Both `Nexus` and the upcycled `MoE (linear router)` consist of 1 shared and 6 routed experts, corresponding to a total number of 1.3B parameters where 605M parameters are activated per input for top-2 routing (1 expert always activated, 1 chosen by the router). The `dense merging` baseline is created by averaging the weights of all dense experts and the seed model, and therefore has the same number of parameters as the seed model.

Compared to the `seed model`, `Nexus` performs better in all evaluation categories with a 5.8% relative gain on average (38.5 vs 36.4). Compared to upcycled models, `Nexus` outperforms `MoE (linear router)` in 3 out of 4 categories with 3.2% relative gain (38.5 vs 37.3) on average, and beats `dense merging` by 8.5% overall relative increase (38.5 vs 35.5). Notably, while both upcycled MoEs outperform the `seed model`, `dense merging` underperforms on average, showing the benefits of MoE upcycling over parameter averaging.

**2.8B parameter seed model.** Next, we experiment by upcycling dense models with 2.7B parameters to validate if the results from the 470M seed model hold at a larger scale. Table 1 compares `Nexus` with `MoE (linear router)` and `dense merging`. Both `Nexus` and `MoE (linear router)` use 1 shared expert and 4 routed experts in these experiments, corresponding to 4.3B active parameters per input (top-2) out of 9.1B total parameters.

Our results show that `Nexus` leads to higher upcycling results compared to the baselines at the 2.8B scale, confirming the findings from smaller scale experiments. `Nexus` enables a 7.4% relative gain over the seed model and outperforms the `MoE (linear router)` with a 1.6% relative increase (50.6 vs. 49.8). `Nexus` outperforms the best baseline in 3 out of 4 task categories and achieves the highest increase in *knowledge* tasks with 22.5% and 5.6% relative to the seed model and the

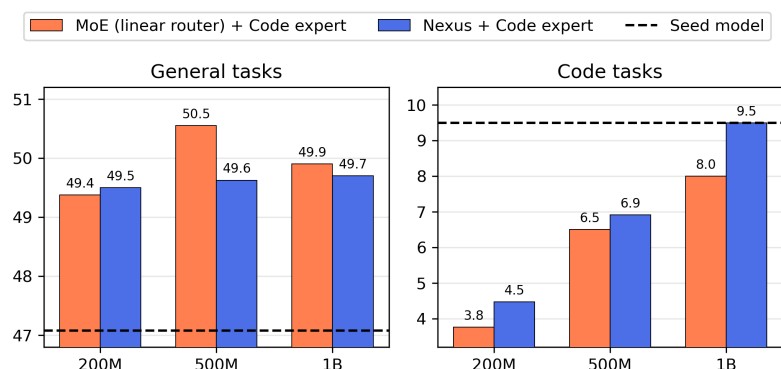

Figure 3: **Extending upcycled MoE models with the Code experts:** After initial upcycling, we extended MoEs (both `Nexus` and `MoE with linear router`) using an independently trained dense Code expert and finetuned the resulting models small number of tokens (200M, 500M, and 1B finetuning tokens) as described in 3. `Nexus` consistently outperforms the baseline in Code performance after extension without losing general performance. General tasks is the macro average of the knowledge, science, reasoning, and general knowledge categories reported in section 5.1. Note that the dense Code expert achieves scores of 42.1 and 14.3 for general and code tasks respectively.

`MoE (linear router)` respectively. These tasks include knowledge retrieval from Wikipedia in which one of our specialized experts is trained for.

Similar to the 470M experiments, both `Nexus` and `MoE (linear router)` outperform the `dense merging` baseline. We relate this to potential cross-task interference between diverse specialized experts (including the seed model as an additional expert), leading to poor performance by applying a simple weight averaging.

## 5.2 EXTENDING THE UPCYCLED MoE MODEL WITH A NEW EXPERT

To support fully modular and efficient training of MoEs, besides upcycling the existing expert models, it is crucial for an adaptive method to have the ability to continuously extend the upcycled MoE with new experts trained using previously unseen data domains. To evaluate this, we train a dense CODE expert and extend the upcycled MoEs (both `Nexus` and `MoE (linear router)`) as described in Section 3. We perform a small-scale finetuning of up to 1B tokens after extending the models. Figure 3 shows both the general performance and the target code performance at 200M, 500M, and 1B finetuning tokens. Here, we ask "*Can we continuously upcycle dense models into an MoE without requiring large-scale MoE training each time?*"

**Performance on the new domain.** As shown in Figure 3 (right), `Nexus` outperforms the `MoE (linear router)` for 200M, 500M and 1B finetuning tokens with 18.4%, 6.2% and 18.8% relative gains respectively. Unlike `MoE (linear router)`, where the router weights are reset after extending the MoE layers, `Nexus` uses the information that is available about the new domain by mapping the domain embedding to a new expert embedding for the router, and therefore finetunes the router weights without a restart.

**Comparison with the dense models.** Nexus reaches the code performance of the seed model while retaining superior performance on general tasks. In comparison to the seed model and the dense code expert (trained for 8B code-only tokens on top of the seed model), although the dense code expert still performs higher than both upcycled MoEs with a score of 14.3, its performance on general tasks is far inferior (42.1). Our method also achieves up to 18.8% relative gains over the `MoE (linear router)`. These results show that with a fraction of the original upcycling budget (1B vs 40B tokens for initial upcycling, and 1B vs 8B tokens for code expert training), `Nexus` can acquire a new capability.

**Performance on general tasks.** As a proxy for the knowledge for previously learned domains, Figure 3 (left) shows the average performance of `Nexus` and `MoE (linear router)` in general tasks. Although there is a slight drop on the general tasks for `Nexus` compared to initial upcycling

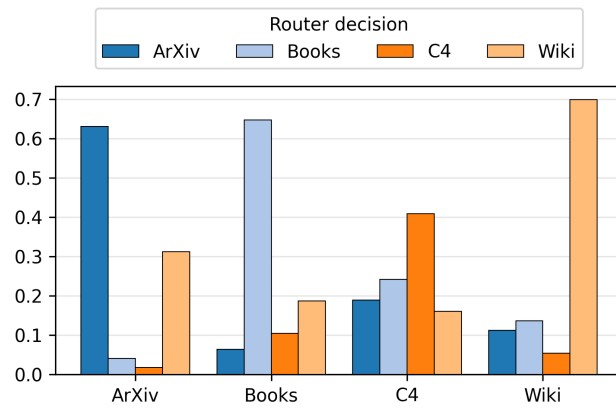

Figure 4: **Average routing probabilities for each expert per domain in Nexus:** We compute the average routing probabilities across Transformer blocks for 512 samples per domain (from the 2.8B experiment). The x-axis denotes the samples' domain and the colored bars show the routing probabilities for the corresponding expert. We show the domains that are used to train specialized experts.

(a relative decrease of 1.9%), the competitive performance is maintained across different numbers of finetuning tokens. We relate this to the composition of the finetuning mix where we use a high percentage of the code data (50% of the code and 50% of the previous domains).

## 5.3 EXPERT SPECIALIZATION

To measure the specialization in our MoE, we take a closer look at how the MoE experts are activated for samples of separate domains. We compute average routing frequencies across all Transformer layers in Figure 4, where the labels on the x-axis represent which domain the tokens are coming from, and the colored bars show the routing frequencies for each of the experts trained on one of the domains. Since we select only one routed expert per token in each MoE layer, and expert FFN layers are inherited from dense experts, average routing frequencies present a good proxy for specialization of each of the experts. Here, we ask "*can Nexus retain a high degree of specialization after upcycling?*"

**Routing for the upcycled experts.** As shown in Figure 4, we find that the expert trained on the corresponding domain always receives the highest share of the tokens from that domain, confirming that Nexus retains the specialization from the specialized dense models. Concretely, this specialization is higher for ArXiv, Books, and Wikipedia with 63.0%, 64.7%, and 69.8% respectively. Interestingly, tokens from C4 are routed only 40.9% of the time to the C4 expert and distributed to the other experts approximately 20% for each one. We relate this to the broad coverage of the C4 dataset, which potentially includes samples closer to other domains and also a large percentage of the C4 used in the MoE training phase (proportional to its size in the SlimPjama dataset). Especially the latter factor pushes tokens from C4 to be distributed to the other experts due to the load balancing factor.

**Specialized routing for the new expert.** Next, we measure expert specialization for the newly added expert on the new code domain. Figure 5 shows the average routing probability per expert for sampled code tokens. We compute routing probabilities on the Nexus model with the code expert after 1B finetuning tokens (See Section 5.2 for details). Here, we see clearly that code tokens are routed to the code expert 69.1% of the time on average. This shows that Nexus not only retains the specialization for the initial upcycling but also exhibits a high degree of specialization for a newly added expert for its own domain.

## 5.4 ABLATIONS

Mixture-of-expert models are known to be sensitive to the choice of load balancing loss factor (Fedus et al., 2022; Zoph et al., 2022) and sampling weights for each data domains during training. As additional ablations, we run two new sets of experiments at 470M scale, one with a lower load balancing factor and the other one with equal weighting of each domain during training (whereas originally the weights were proportional to the share of tokens of that domain in SlimPajama). Figure 6 compares Nexus and MoE(linear router) in terms of their downstream performances for these ablations. Finally, in this section, we also visualize domain and projected expert embeddings to see if the relationship between embeddings is preserved after the learned projection.

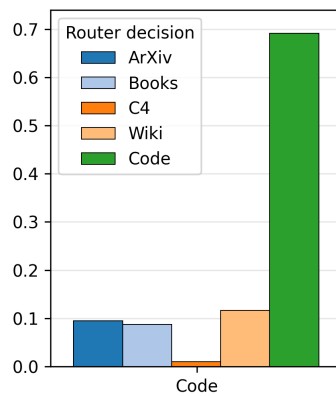

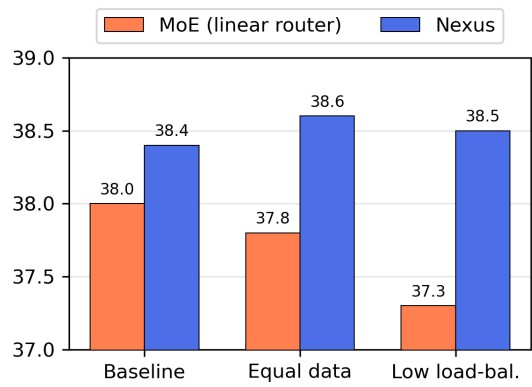

Figure 5: **Average routing probabilities per expert for the new domain in extended Nexus:** We show the routing probabilities for code tokens after extending MoE (1B finetuning).

Figure 6: **Comparison between Nexus and the baseline in different load balancing and data sampling setups:** We compare Nexus and MoE (linear router) by lowering the load balancing loss factor and uniformly sampling the data domain during training in isolation.[3]

**Lowering the load balancing loss factor.** In Figure 6 (baseline vs low load-bal.), we compare two `Nexus` models with the corresponding `MoE (linear router)` baselines where we use load balancing loss factor of 0.05 and 0.0005 for each set of experiments. We find that using a significantly lower factor for the load balancing loss hurts `MoE (linear router)` performance by approximately 2% relative drop while `Nexus` shows a robust performance across both load balancing factors. We hypothesize that because the expert embeddings in our router are always based on the domain representations, we achieve more stable distribution of tokens even if the load balancing loss is weighted extremely low.

**Changing the training data composition.** Next, we compare our default of sampling specialized domain data proportional to the size of the domain (total amount of tokens in SlimPajama), with a uniform sampling over all domains. Figure 6 (baseline vs equal data) shows the downstream performances for both `Nexus` and `MoE (linear router)`. Although sampling uniform sampling domains' data does not significantly impact the downstream performance for both models, we find that it helps `Nexus` to improve specialization for all the domains in terms of expert routing probabilities (Figure 11, Appendix I). In particular, compared to the size proportional sampling, tokens from the C4 domain are routed more accurately (27.6% vs 71.1%) when data is equally sampled.

**Domain embeddings before and after projection.** Finally, in Figure 8, we visualize cosine similarities between domains and the projected expert embeddings from the last Transformer block, in our main upcycling experiments at the 470M scale. Comparing the embeddings before and after mapping, we find that the router's learned projection preserves the main relationship between domains. For instance, relatively high cosine similarity between Books & C4, and StackExchange & GitHub exist both between their domain embeddings and the projected expert embeddings. Interestingly, while preserving the main relationships, we also find that the learned projection pushes expert embeddings further away from each other, potentially due to our choice of only activating a single expert per token besides the shared expert.

# 6 RELATED WORK

**Routing Variants of MoEs.** The most common MoE architecture (Shazeer et al., 2017; Lepikhin et al., 2020; Fedus et al., 2022) employs a linear router with a top-$k$ routing scheme, where $k$ typically equals 1 or 2. In this standard routing schema, only the $k$ experts with the highest router gate values are activated. There is substantial research proposing alternatives to top-$k$ expert assignments (Hazimeh et al., 2021; Lewis et al., 2021; Roller et al., 2021; Zhou et al., 2022; Zuo et al., 2022). DeepSeek-MoE (Dai et al., 2024) introduces a routing variant where a number of experts are "*shared*"

---

[3]We report the average performance on Knowledge, Science, Reasoning, and MMLU.

and always assigned to all tokens. Our work also adopts this approach for our general base expert. However, these efforts primarily focus on improving the general performance and/or training stability of MoEs. In contrast, our work puts emphasis adaptability and extensibility.

**Efficient MoE Training by Re-Using Existing Dense Models.** Training MoEs from scratch is computationally expensive (Gale et al., 2023; Fedus et al., 2022) and often challenging due to training instabilities (Zoph et al., 2022). Alternatively, recent works have explored re-using existing dense models to initialize MoEs. Sparse Upcycling (Komatsuzaki et al., 2023) re-uses a single dense model to initialize the MoE by replicating the FFN weights in an MoE layer. The router is initialized randomly, and all other parameters are copied directly from the dense model. BTX (Sukhbaatar et al., 2024) extends this approach by upcycling not from a single dense model, but from multiple specialized dense expert models. Furthermore, BAM (Zhang et al., 2024) expands BTX to upcycle not only FFN experts but also attention experts. Our work also leverages this approach by reusing specialized dense experts for an MoE, while extending it further to facilitate on-the-fly adaptations for new experts specialized in unseen data domains.

**Efficient MoE Architectures.** Zadouri et al. (2024) proposes replacing traditional MoE's computation-heavy feed-forward network (FFN) experts with more efficient experts comprised of smaller vectors and adapters, which are activated in parallel to a single dense FFN. This lightweight architecture necessitates only a limited number of parameter updates when finetuning, offering efficiency advantages. However, unlike our approach, it does not leverage existing specialized dense models and lacks a notion of specialized experts, which are central to our method. Similar to our work, Muqeeth et al. (2024) and Ostapenko et al. (2024) study combining separately trained experts into a unified model. However, they focus on parameter-efficient adapters such as LoRA (Hu et al., 2021) and supervised finetuning. In this work, we focus on efficiently pre-training fully-fledged MoE models via upcycling.

**Adaptive MoEs and Ensemble Models.** ModuleFormer (Shen et al., 2023) also aims to produce adaptive MoEs. The authors achieve adaptability by freezing existing MoE parameters while only training newly added modules with optimization constraints to the router. Unlike our work, ModuleFormer does not leverage existing expert dense seed models for efficiency gains, nor does it have a notion of specialization which is central to our work. Similar to our work, DEMix (Gururangan et al., 2021) independently trains different FFN experts on specialized data domains, with each expert functioning as a domain-specific module. Modules can be added on-the-fly for adaptability. Followup works BTM and C-BTM (Li et al., 2022; Gururangan et al., 2023) extend DEMix to create adaptive ensemble models. However, all three works use a router requiring a forward pass for every expert at inference instead of sparsely activating them, which significantly increases inference costs, especially with a large number of experts. Unlike these approaches, our router cost is approximately the same as standard top-$k$ routing during both training and inference, offering a more scalable solution for adaptability.

## 7 CONCLUSION

We propose `Nexus`, a new LLM framework that enables efficient upcycling of specialized dense experts into a sparsely activated MoE model. We show that individual experts in our method retain their specialization after upcycling, and that our router based on expert embeddings outperforms previous approaches for combining the dense experts. Furthermore, the model can be extended efficiently with new dense experts after the initial training phase, saving much compute compared to re-training the upcycled model or training from scratch.

## 8 LIMITATIONS

The MoE architecture is often employed for larger models in the multi-billion parameter range, where efficiency is paramount. However, to facilitate a broader set of experiments, we limit our setup to using 2.8B parameter seed models for the main results and 470M parameter seed models for ablations. Furthermore, our dense experts are based on existing data sources in the SlimPajama dataset which is pre-defined. Future work could extend our method by discovering specialized data domains through unsupervised clustering similar to Gururangan et al. (2023).

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

## A  NEXUS ROUTING ALGORITHM

Figure 7 outlines the code for the Nexus router, which consists of **(1)** a 2-layer MLP network (`domain_to_expert_ffn`) to project domain embeddings to expert embeddings, **(2)** shared and routed expert FFNs, and **(3)** sparse Top-k gating. Note that the expert embeddings are independent of the input and could be precomputed once and stored as long as the weights of the model do not change. This means that the routing layer during inference closely resembles a vanilla MoE router, with the difference being that the router matrix in Nexus is not learnt during training but computed using the domain embeddings as an informative prior.

```python
def router(self, inputs, domain_embeddings):
    # domain_to_expert_ffn learns projection domain to expert embeddings
    # domain_embeddings: [e_dim x n_experts]
    # expert_embeddings: [h_dim x n_experts]
    expert_embeddings = self.domain_to_expert_ffn(self.domain_embeddings)

    # router probs: [batch, seq, n_experts]
    router_probs = nn.softmax(inputs @ expert_embeddings)

    # Top-1 gate for routed experts
    index, gate = nn.topk(1, router_probs)

    # routed_experts_ffns: An MoE layer with FFN experts
    # routed_expert_out: [batch, seq, h_dim]
    # shared_expert_out: [batch, seq, h_dim]
    routed_expert_out = self.routed_expert_ffns[index](input)
    shared_expert_out = self.shared_expert_ffn(input)

    return shared_expert_out + gate * routed_expert_out
```

Figure 7: **Router layer in Nexus:** PyTorch-like pseudo-code illustrating the routing mechanism, situated before the expertized MLP layer in each transformer block.

## B  COMPARISON OF EXISTING APPROACHES WITH NEXUS

Table 2 compares Nexus with previous approaches in the field of efficient MoE training. Unlike the vanilla MoE architecture (Shazeer et al., 2017; Fedus et al., 2022), the Branch-Train-Merge (BTM; Li et al., 2022) and the Branch-Train-Mix (BTX; Sukhbaatar et al., 2024) approaches train experts separately in different domains, reducing training cost and improving specialization. However, they either merge the experts during inference (BTM) or learn an MoE router layer from scratch, where prior domain information is not used (BTX). Our approach trains the MoE router based on domain information, maintaining the specialization and enabling efficient extension of the MoE with a new expert after training.

| | MoE (Vanilla) | BTM (Merge) | BTX (Linear router) | **NEXUS** (Ours) |
|---|:---:|:---:|:---:|:---:|
| Dense experts are trained independently (upcycling) | ✗ | ✔ | ✔ | ✔ |
| Experts are specialized in different domains | ✗ | ✔ | ✔ | ✔ |
| Experts are chosen by a learned router per input token | ✔ | ✗ | ✔ | ✔ |
| Router is adaptive via learned projection for new domains | ✗ | ✗ | ✗ | ✔ |

Table 2: **A comparison of existing approaches with Nexus:** We choose the vanilla MoE architecture (Shazeer et al., 2017; Fedus et al., 2022), Branch-Train-Merge (BTM; Li et al., 2022), and Branch-Train-Mix (BTX; Sukhbaatar et al., 2024) for comparison. Nexus combines the advantages of the existing MoE extensions while also allowing easy adaptation to new domains.

Furthermore, Table 3 shows parameter counts during training and inference of Nexus vs. the baselines. From this, we can infer that Nexus has the same memory and compute complexity as a vanilla MoE model during inference, and a slight overhead of $\tilde{1}$% additional trainable parameters during training.

| | Total Parameters | Active Parameters (Training) | Active Parameters (Inference) |
|---|---|---|---|
| **470M Models** | | | |
| SEED MODEL (470M) | 467,682,304 | 467,682,304 | 467,682,304 |
| MOE (LINEAR ROUTING) | 1,298,252,800 | 606,110,720 | 606,110,720 |
| NEXUS | 1,312,834,560 | 620,692,480 | 606,110,720 |
| **2.8B Models** | | | |
| SEED MODEL | 2,752,565,760 | 2,752,565,760 | 2,752,565,760 |
| MOE (LINEAR ROUTING) | 9,044,226,560 | 4,325,429,760 | 4,325,429,760 |
| NEXUS | 9,129,218,560 | 4,410,421,760 | 4,325,429,760 |

Table 3: **Total and active parameter counts.** Comparison of the seed model, linear MoE, and Nexus architectures for both 470M and 2.8B parameter models. During inference, the router weights of Nexus can be precomputed once by the learned MLP hypernetworks, making it exactly equal to the vanilla MoE in terms of memory and compute complexity. During training, we also observe exactly the same step time for the vanilla MoE and Nexus, as the overhead of the additional MLP is negligible. In the 470M category, the MoE/Nexus models use 6 routed and 1 shared expert. In the 2.8B category, the MoE/Nexus models use 4 routed and 1 shared expert. In both categories, the models activate the shared expert and the top-1 of the routed experts during inference.

## C   EVALUATION DETAILS

For the downstream evaluation, we measure the performance of each model on 15 tasks[4] from five evaluation categories that reflect different capabilities based on the tasks and the datasets used in the benchmarks:

- **Knowledge**: To measure question-answering capabilities based on world knowledge and web documents such as Wikipedia, we report the performance on OpenBookQA (Mihaylov et al., 2018), Natural Questions (Kwiatkowski et al., 2019), TriviaQA (Joshi et al., 2017), QUAC (Choi et al., 2018) (all 0-shot) and SQuAD (4-shot) (Rajpurkar et al., 2016).

- **Science**: For measuring knowledge in science-oriented academic benchmarks, we use ARC-Easy, ARC-Challenge (Clark et al., 2018), SciQ (Welbl et al., 2017) (all 0-shot).

- **Reasoning**: For reasoning abilities, we use CommonSenseQA (Talmor et al., 2019), SIQA (Sap et al., 2019), PIQA (Bisk et al., 2020), WinoGrande (Sakaguchi et al., 2019), and HellaSwag (Zellers et al., 2019) (all 0-shot).

- **General Language Understanding**: We use MMLU (5-shot) (Hendrycks et al., 2021) to test general language understanding.

- **Code**: For code generation, we evaluate models on MBPP (Austin et al., 2021), LBPP (Matton et al., 2024), and HumanEval-Pack (Chen et al., 2021) which includes Cpp, Javascript, Java, Go, Python, and Rust (all 0-shot).

## D   RESULTS FOR THE 470M PARAMETER MODEL

Table 4 shows the downstream task results for Nexus with a 470M parameter seed model. Our approach outperforms the baselines in all downstream benchmarks. `Dense merging` corresponds a dense model with 470M parameters, while both `Nexus` and `MoE (linear router)` consist of 605M active and 1.3B total parameters.

---

[4]We did not include ARC-Challenge and Natural Questions in 470M experiments as some model variants were unable to achieve non-random performance.

|                        | Know. | Science | Reason. | MMLU | Avg. |
|------------------------|-------|---------|---------|------|------|
| SEED MODEL (470M)      | 14.0  | 51.4    | 50.5    | 29.8 | 36.4 |
| **Upcycled Models**    |       |         |         |      |      |
| DENSE MERGING          | 10.9  | 52.0    | 50.3    | 27.8 | 35.5 |
| MoE (LINEAR ROUTER)    | 13.4  | **55.0**| 51.3    | 29.6 | 37.3 |
| NEXUS                  | **16.7**| **55.0**| **52.3**| 29.8 | **38.5** |

Table 4: **Downstream task results for Nexus with a 470M parameter seed model.** `Dense merging` merges all separately pretrained experts, while both `Nexus` and `MoE (linear router)` upcycle them and are evaluated with top-2 routing.

## E  RESULTS FOR INDIVIDUAL EXPERTS

To further contextualize the performance of the Nexus models, we report the performance of each individual expert in Table 5. The experts initialized from the 470M seed model are trained for 20B tokens on their domains, while the experts initialized from the 2.8B seed model are trained for 40B tokens.

|                       | Know. | Science | Reason. | MMLU | Avg. |
|-----------------------|-------|---------|---------|------|------|
| **470M Experts**      |       |         |         |      |      |
| ARXIV                 | 9.5   | 47.8    | 44.3    | 31.2 | 33.2 |
| BOOKS                 | 9.0   | 51.8    | 51.4    | 32.0 | 36.1 |
| C4                    | 3.9   | 52.6    | 51.5    | 27.6 | 33.9 |
| GITHUB                | 11.3  | 44.8    | 45.2    | 30.2 | 32.9 |
| STACKEXCHANGE         | 9.9   | 45.4    | 44.9    | 29.2 | 32.4 |
| WIKIPEDIA             | 15.3  | 46.4    | 44.1    | 25.4 | 32.8 |
| **2.8B Experts**      |       |         |         |      |      |
| ARXIV                 | 13.4  | 57.3    | 51.3    | 36.2 | 39.5 |
| BOOKS                 | 19.4  | 62.5    | 60.0    | **39.6** | 45.4 |
| C4                    | 11.0  | 64.5    | 61.9    | 37.8 | 43.8 |
| WIKIPEDIA             | 22.6  | 60.3    | 55.3    | 37.2 | 43.9 |
| CODE                  | 13.4  | 59.9    | 52.4    | 37.8 | 40.9 |
| **Upcycled Models**   |       |         |         |      |      |
| NEXUS (470M)          | 16.7  | 55.0    | 52.3    | 29.8 | 38.5 |
| NEXUS (2.8B)          | **33.2**| **67.3**| **62.6**| 39.4 | **50.6** |

Table 5: **Downstream task performance of individual experts.** We report the separate performance of all experts used during the upcycling and extension stages. Note that the Nexus models beat every individual expert used for their upcycling, with one exception.

## F  RESULTS FOR CONTINUAL TRAINING OF THE SEED MODEL

To compare Nexus to another dense baseline, for Table 6 we continually train the 470M and 2.8B seed models in a data matched setting. This means the 470M model has seen a total of 750B pretraining tokens (general pretraining data mix), 120B tokens from SlimPajama domains (the shuffled training tokens of all 6 experts), and 25B tokens from SlimPajama to match the Nexus finetuning phase. The 2.8B model has seen a total of 750B pretraining tokens, 160B tokens from SlimPajama (the shuffled training tokens of all 4 experts), and 40B additional tokens from SlimPajama to match the Nexus finetuning phase.

|  | Know. | Science | Reason. | MMLU | Avg. |
|---|---|---|---|---|---|
| **470M Models** | | | | | |
| SEED MODEL | 14.0 | 51.4 | 50.5 | 29.8 | 36.4 |
| SEED MODEL + 145B TOKENS | **19.9** | 53.8 | 50.8 | 29.6 | **38.5** |
| NEXUS | 16.7 | **55.0** | **52.3** | 29.8 | 38.5 |
| **2.8B Models** | | | | | |
| SEED MODEL | 27.1 | 62.0 | **63.8** | 35.4 | 47.1 |
| SEED MODEL + 200B TOKENS | 28.8 | 66.4 | 62.7 | 41.4 | 49.8 |
| NEXUS | **33.2** | **67.3** | 62.6 | **39.4** | **50.6** |

Table 6: **Downstream task results for data-matched continued pretraining of the 470M and 2.8B seed models.** Both seed models are data matched to the `Nexus`/`Linear MoE` variants, including all expert training and finetuning. For the 2.8B parameter models, we also train for 1B tokens on instruction-style datasets from the original pretraining data before measuring the performance on downstream tasks (see Section 4.2). Note that the seed model training takes a lot more wallclock-time compared to our method, as the Nexus experts can all be trained in parallel, which is not possible with a single model.

## G    COMPARISON OF DOMAIN EMBEDDINGS AND EXPERT EMBEDDINGS

Nexus maps the *domain embeddings* which are computed from each domain's training dataset to *expert embeddings* which represent experts. Figure 8 shows that trends in the similarity matrix are preserved after the mapping, but the embeddings are pushed away from each other a bit (lower similarity).

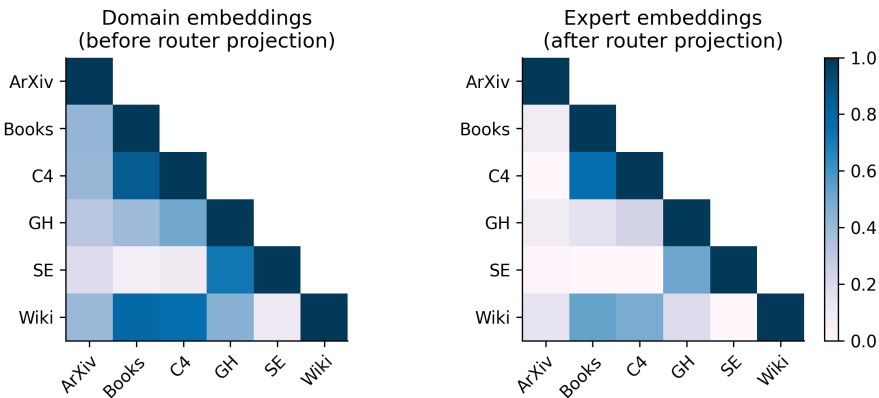

Figure 8: **Domain and the projected expert embeddings for Nexus:** We visualize cosine similarities between domains and the projected expert embeddings from the last Transformer block. The similarities are obtained from the 470M experiments. Our projected router maintains the relative similarity between the original domains (e.g. Books & C4, Github & StackExchange) after the router's projection.

## H    ROUTING PROBABILITIES FOR THE LINEAR MOE MODEL

To investigate how specialized individual experts are in the `Nexus` approach vs. the vanilla MoE baseline, we also compute the routing distributions for the `MoE (Linear Router)` baseline with 2.8B parameters. Figure 9 shows the router distribution of this model with 4 experts. Figure 10 shows the router distribution for code data after adding the new code expert to the baseline. Although the specialization of the linear MoE model (Figure 9) matches that of Nexus for pretraining (Figure 4), it adapts much worse to the new expert, as fewer tokens from the code domain actually get routed to the code expert (Figure 10) than with Nexus (Figure 5).

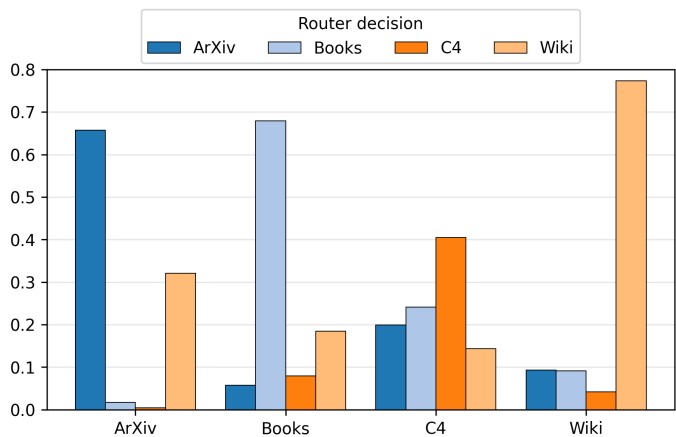

Figure 9: **Average routing probabilities for each expert per domain in the `MoE (Linear router)` baseline:** We compute the average routing probabilities across Transformer blocks for 512 samples per domain (from the 2.8B experiment). The x-axis denotes the samples' domain and the colored bars show the routing probabilities for the corresponding expert. We show the domains that are used to train specialized experts.

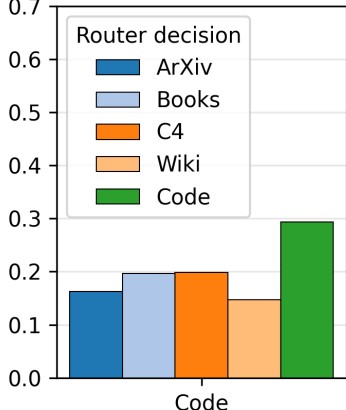

Figure 10: **Average routing probabilities per expert for the new domain in the extended `MoE (Linear router)` baseline:** We show the routing probabilities for code tokens after extending the MoE with a new expert and finetuning for 1B tokens.

## I    ROUTING PROBABILITIES FOR UPCYCLING ABLATIONS

Figure 11 shows the expert routing probabilities for Nexus for all three settings described in Section 5.4.

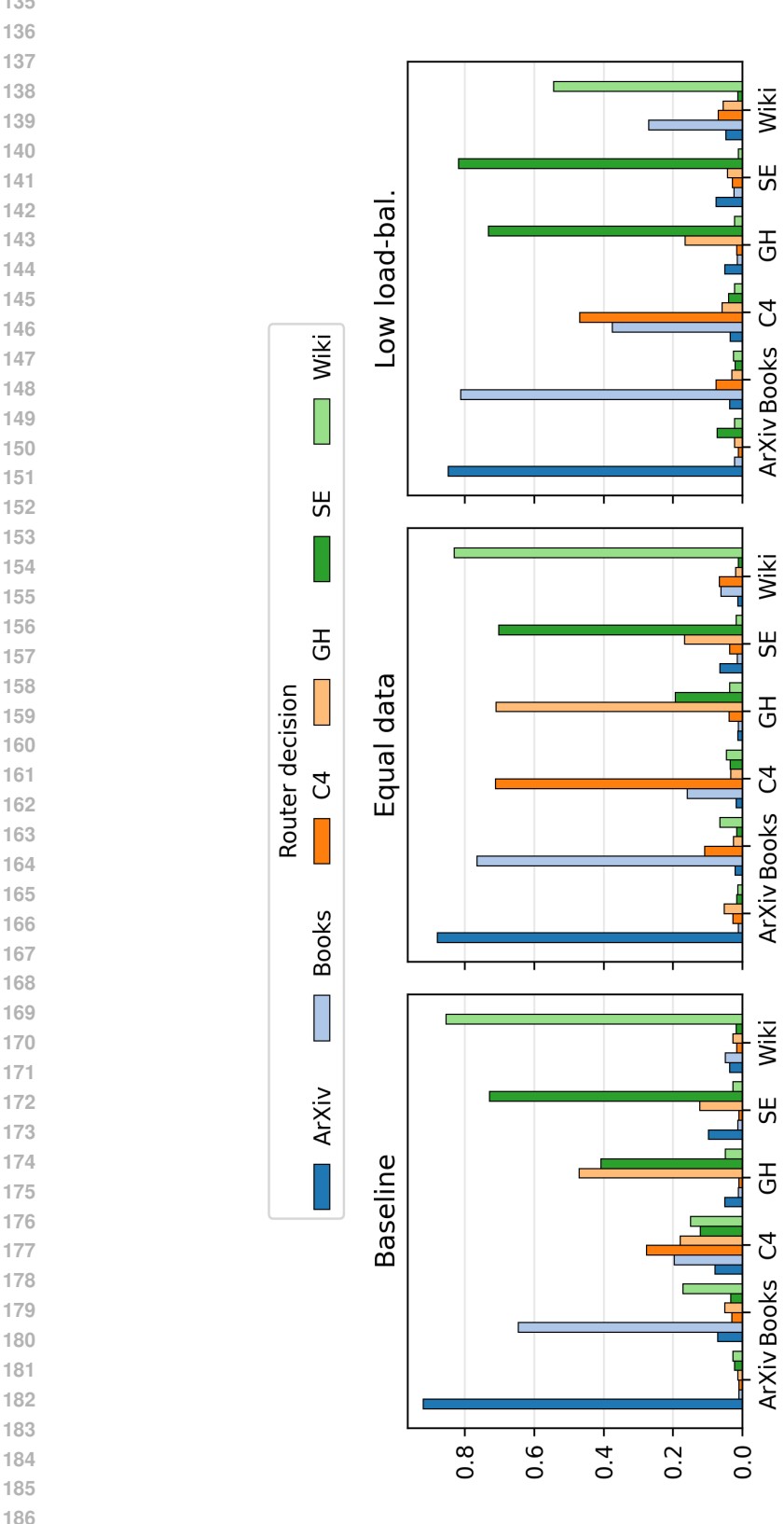

Figure 11: **Average routing probabilities for each expert per domain in different upcycling setting:** We show expert routing probabilities for Nexus for all three settings described in Section 5.4.

