# OpenReview forum: "Nexus: Specialization meets Adaptability for Efficiently Training Mixture of Experts"
_ICLR.cc/2025/Conference — Submitted to ICLR 2025_

### Official Review · Reviewer_fKKY · 2024-10-18

**Soundness:** 2
**Presentation:** 2
**Contribution:** 2
**Rating:** 3
**Confidence:** 4

**Summary:**

This paper provides a MoE framework called Nexus that improves efficiency, specialization, and adaptability.

**Strengths:**

1) Efficiency, specialization, and adaptability are important when training a MoE network.
2) The paper provides a novel router based on expert embeddings that outperform previous approaches for combining dense experts.
3) The paper provides detailed experiments on the designed methods.

**Weaknesses:**

1) Lack of technical contributions:
- The methodology part of the paper is too short.
- Some methods mentioned in the paper have been explored in previous research. For example, “shared expert” has been discussed in Deepseekmoe [1], and the adaptation to new domains (expert merging) has been discussed in Mergekit [2].

2) Lack of comparison baselines:
This paper lacks baseline models. There are already many open-sourced MoE networks for comparison, e.g., Mixtral [3], Deepseekmoe [1]. The author should compare these models, in terms of training time, data, performance, etc.

[1] Dai D, Deng C, Zhao C, et al. Deepseekmoe: Towards ultimate expert specialization in mixture-of-experts language models[J]. arXiv preprint arXiv:2401.06066, 2024.

[2] https://github.com/arcee-ai/mergekit

[3] Jiang A Q, Sablayrolles A, Roux A, et al. Mixtral of experts[J]. arXiv preprint arXiv:2401.04088, 2024.

**Questions:**

As in Weaknesses.

---

> ### Author Response · Authors · 2024-11-21
> **Author reply**
>
> We would like to thank R fKKY for their time and the feedback provided. We appreciate any critical insights and will take them into account as we continue refining our work.
>
> We want to address the points raised one by one:
>
> > **The methodology part of the paper is too short.**
>
> We provide a comprehensive description of our method in “Background” (Section 2), and “Adaptive Router for Upcycling Specialized Experts as MoE” (Section 3). Additionally, we also provided pseudo code for the router showing its details in Appendix A. Together with the method, we also provide very detailed explanation of our experiments in Section 4 which helps the reader understand our method. We believe that these sections clearly describe our methodology and demonstrate our experimental setup.
> We are happy to extend any missing point or unclear description if *R fKKY* points out so. We kindly would like to note that length of the methodology sections should not be considered as a criteria for rejecting an ICLR paper.
>
> > **Some methods mentioned in the paper have been explored in previous research. For example, “shared expert” has been discussed in Deepseekmoe, and the adaptation to new domains (expert merging) has been discussed in Mergekit.**
>
> Thanks for your comments. We would like to clarify that our paper’s main novel contribution is the novel router that leverages information about the training domain of specialized experts to enable better MoE upcycling from separately trained dense models.
>
> For the use of shared experts, we already cite the original paper where shared experts are trained end-to-end which is different from our setting. It is worth mentioning that the use of shared experts is new in our upcycling setting. Previous work such as BTX [1], used seed models MLP’s as separate experts but does not utilize it as a shared expert. In our experiments, we find that using the seed model’s MLP as the shared expert is particularly beneficial when adapting a new domain expert into Nexus (i.e. extending nexus with new domain expert). We believe that this is novel and demonstrates a new approach compared to the baseline in upcycling setting.
>
> For the merging of the attention weights, we follow BTX [1] by simply averaging parameters and clearly mention this in the manuscript. We are happy to cite Mergekit as well in the related works in the context of merging.
>
> [1] Sukhbaatar, S., Golovneva, O., Sharma, V., Xu, H., Lin, X. V., Rozière, B., Kahn, J., Li, D., tau Wen-Yih, Weston, J., & Li, X. (2024). Branch-Train-MiX: Mixing Expert LLMs into a Mixture-of-Experts LLM. https://arxiv.org/abs/2403.07816
>
> > **Lack of comparison baselines: This paper lacks baseline models. There are already many open-sourced MoE networks for comparison, e.g., Mixtral [3], Deepseekmoe [1]. The author should compare these models, in terms of training time, data, performance, etc.**
>
> Thanks to the suggestions by R sqxx and R Ykss, we extended the baseline comparisons with data-matched seed models, where we show Nexus outperforms dense baselines even when they are further trained much longer with the data of all the experts.
>
> However, comparing Nexus with Mixtral [2], or DeepseekMoE [3] would not be fair due to multiple reasons:
> * Both models are much larger than the models that we trained in this paper. Mixtral includes 46.7B and DeepseekMoE has 16.4B total parameters.
> * DeepseekMoE is trained for 2T tokens which is a much larger token count compared to our experiments. Also, they do not disclose the detailed information for the training datasets. Similarly, Mixtral does not disclose any information about their training data and number of training tokens.
>
> Therefore, we believe that comparing models trained in this paper with these MoEs would not be fair.
>
> As the main baseline, we compared Nexus with the BTX [4]-style standard MoE models that we upcycle from the same dense experts, using the exact same dataset. In this way, we aimed to isolate the benefits of our proposed router and showcase that Nexus router outperforms the linear router for upcycling MoE from specialized experts without any additional cost. In particular, when extending an upcycled MoE with a new domain, our experiments (Section 5.2, Figure 3) show that Nexus enables clearly better performance for the new domain with a lightweight fine-tuning.
>
> As we address their comments, we would like to ask R fKKY to consider re-evaluating their review.

---

> ### Author Response · Authors · 2024-11-23
> **Follow-up**
>
> As the discussion period is nearing its end, we wanted to ask **R fKKY** if there are any follow-up points we can clarify. We have responded to all concerns raised, in addition to taking a few days to run the additional experiments necessary to demonstrate our points, all of which are incorporated in the appendix of the manuscript. If there are no further points of clarification regarding the manuscript, our methodology, our extended baselines, and the difference between our method and existing expert sharing and merging approaches, we would kindly ask that reviewer **R fKKY** considers increasing their score to reflect this.

---

### Official Review · Reviewer_sqxx · 2024-10-31

**Soundness:** 3
**Presentation:** 3
**Contribution:** 2
**Rating:** 5
**Confidence:** 4

**Summary:**

This work propose Nexus, a strategy for upcycling dense LLM checkpoints into SMoE while enhancing the experts specialization and adaptation to new domains. Nexus works by first independently train a seed LLM on each domain and then upcycle them via a domain-aware router. The authors compare Nexus against other upcycling strategies in the language modeling task. The results show that Nexus can offer some performance gains while improving the specialization and adaptability.

**Strengths:**

- SMoE is an important and interesting research direction. This work shows that improving experts specialization can be helpful and can generalize the new domains.
- The experiments involve training LLMs from scratch at a reasonable scale.
- The method is simple and intuitive sound. The results are encouraging and can corroborate the motivation.

**Weaknesses:**

- One major drawback of this work is that Nexus introduces a dependency on the domain associated to each sample, which does not exist in the baselines. Conceptually, I think this is not a Wdesirable property for real-world scenarios as it requires additional data annotation during deployment. Moreover, training and inference could be more susceptible to data mislabeling or low quality-data. Furthermore, this also introduces an advantage of Nexus over the baselines since it receives an additional input signal. I believe this dependency is an important aspect that is not fully explored in this work.

- The paper does not include a complexity analysis.

**Questions:**

**Questions for the authors**

- How would the dependency on the domain impact the performance and practicality of this work?
- What are the training, inference, and memory costs of Nexus compared to the baselines?
- Comparison with the seed model is not entirely fair as it is only trained on the SlimPajama dataset and not on the domain specific data. Two baselines for further comparison could be: (i) continue training the seed model on a mix of all these domain data; and (ii) maintaining a collection of all specialized experts and select the corresponding one for inference based on the domain index, this can be considered as the fully specialized baseline.
- Will the authors make the data and source code publicly available for reproducibility and further research?

---

> ### Author Response · Authors · 2024-11-21
> **Author reply**
>
> We would like to thank R sqxx for the detailed and thoughtful feedback. We appreciate that the reviewer acknowledges that “This work shows that improving experts specialization can be helpful and can generalize the new domains.”
>
> We want to address the points raised one by one:
>
> > **Nexus introduces a dependency on the domain associated to each sample, which does not exist in the baselines**
>
> We politely want to point out that there seems to be a misunderstanding of the behavior of Nexus during inference.
>
> * Nexus does not require data annotation during deployment, instead it utilizes a learned router to choose which expert to activate for which input.
> * The router weights are calculated by a shallow MLP hypernetwork. The inputs to this network are the embeddings of each of the domains (not only the domain corresponding to the input sample!). The embeddings are constant during the whole training and inference process, and are computed before the training starts by embedding a representative sample of each training data fold once.
> * At inference, not only the inputs to the MLP, but also the weights of the MLP are constant, which also leads to constant outputs. These can be precomputed and hardcoded into the router layer, which makes Nexus during inference absolutely identical to a corresponding vanilla MoE model in terms of architecture and number of total/active parameters. The distinguishing feature is that if an additional expert is added, the Nexus hypernetwork can compute the routing weights for this additional expert, instead of learning it from scratch.
> * This means that Nexus does not have an advantage over the baseline and does not receive an additional input signal.
>
> As mentioned in the manuscript, the domain labels are only required during the training time, however, this labeling can also be simply done through an unsupervised domain clustering similar to c-BTM [1]. We think that assuming the availability of the labels in training time is realistic since large pre-training datasets either include data sources or can be pre-classified for such labels.
>
> In case we misunderstood the intention of this point raised by the reviewer, we politely ask for clarification on this issue.
>
> [1] Gururangan, S., Li, M., Lewis, M., Shi, W., Althoff, T., Smith, N. A., & Zettlemoyer, L. (2023). Scaling Expert Language Models with Unsupervised Domain Discovery. https://arxiv.org/abs/2303.14177
>
> > **What are the training, inference, and memory costs of Nexus compared to the baselines? The paper does not include a complexity analysis.**
>
> We thank reviewer R sqxx for suggesting this addition, and have added Table b) to the global response (also in the new Appendix B, Table 3), comparing the number of total and activate parameters for Nexus and all baselines, which allows comparing the memory and compute cost during training and inference.
>
> At 470M scale, both Nexus and the baseline MoE with linear router include approximately 1.3B total parameters, however, they only use 606M active parameters at inference time. During the training, Nexus additionally updates 15M additional parameters, which correspond to the parameters of the hypernetwork to learn router weights from the domain embeddings. After the hypernetwork is trained, its constant output (as the domain embedding inputs are constant) can be calculated once and stored, afterwards these additional parameters have zero impact on the memory and compute cost of Nexus.
>
> Similarly, at 2.8B scale, Nexus and the baseline have around 9B total and 4.3B active parameters, and Nexus updates 85M additional parameters for the router hypernetwork during training, which have no influence on the inference behavior.
>
> > **Comparison with the seed model is not entirely fair as it is only trained on the SlimPajama dataset and not on the domain specific data.**
>
> We thank reviewer R sqxx for raising this valid point. We have ran the proposed additional baselines and included the results in Table a) in the global response (also in the new Appendix F, Table 6 in the paper). The experiments show that the 2.8B Nexus model outperforms the continued pretraining baseline clearly, while also providing the benefits of extensibility with new experts for easy adaptation to new domains. For more details, we refer to the global response.
>
> The second new baseline proposed by the reviewer, which uses an “oracle model” to always pick the correct expert, is an interesting idea, but unfortunately would require information about the correct domain for a token which is not available to Nexus. While the training data does have domain labels which could be used, this approach would make inference on other datasets impossible, e.g. on downstream task evaluations or in a chat setting.

---

> > ### Author Response · Authors · 2024-11-21
> > **Author reply (continuation)**
> >
> > > **Will the authors make the data and source code publicly available for reproducibility and further research?**
> >
> > We chose to perform all experiments with the open source SlimPajama dataset to improve reproducibility, however, the models were trained on a proprietary codebase and can therefore not be published directly, but we include all necessary details to fully reproduce them, especially for the Nexus router layer. Pseudocode for the Nexus router can be found in Appendix A. We are happy to provide any further information if needed.
> >
> > We sincerely thank the reviewer for their valuable feedback, which has helped us improve the quality of our work.

---

> ### Author Response · Authors · 2024-11-23
> **Follow-up**
>
> As the discussion period is nearing its end, we wanted to ask **R sqxx** if there are any follow-up points we can clarify. We have responded to all concerns raised, in addition to taking a few days to run the extensive experiments necessary to demonstrate our points, all of which are incorporated in the appendix of the manuscript. If there are no further points of clarification regarding the manuscript, domain dependencies, seed model baselines, or the complexity of Nexus compared to the baselines, we would kindly ask reviewer **sqxx** to consider increasing their score to reflect this.

---

> > ### Comment · Reviewer_sqxx · 2024-11-23
> > **Official Comment by Reviewer sqxx**
> >
> > I appreciate the authors' efforts in addressing my concerns. I am satisfied with the authors responses of most questions, except for my concerns regarding the domain embedding.
> >
> > First, I thank the authors for clarifying the training and inference steps of Nexus. I have re-read the paper and still not convinced that "Nexus does not have an advantage over the baseline and does not receive an additional input signal."
> > To me, it is very clear that the key technical contribution of this work is the introduction of the domain information to the router, which improves the results and other nice characteristics as authors claimed. Given this, I believe this domain embedding needs to be investigated more comprehensively; some suggestions are the usage of different embedding models, the sensitivity to mislabeling, or the effect of different labeling techniques. The authors suggested c-BTM, however this work seems to be under review and it is not guarantee to provide useful domain information for SMoE training. Thus, I also agree with fKKY that the technical contribution of this work is rather shallow, i.e.,  it is not very clear to researchers to adopt Nexus to their work due to many uncertainties regarding the domain embedding designs. Together with the fact that the authors will not publish a  fully usage code base to facilitate future research, I am hesitate to support to accept this work in its current form.

---

> > > ### Author Response · Authors · 2024-11-25
> > >
> > > We appreciate that **R sqxx** took the time to review our approach again, and are glad that our response could satisfyingly answer most of their questions.
> > >
> > > We would like to clarify the following points further:
> > >
> > > > **I [am] still not convinced that Nexus does not have an advantage over the baseline and does not receive an additional input signal.**
> > >
> > > We are happy to further elaborate on this point, and clarify the differences and similarities between our architecture and the baselines.
> > >
> > > First, considering the training setting, where Nexus uses the expert training data to speed up router training:
> > > * We split the training data into four domains and train four expert models, each on one domain.
> > > * We embed a representative sample of each expert’s training data **once** with a state-of-the-art embedding model, and **average** the embeddings of each document, which gives us four immutable vectors of dimension d_embed. This embedding step only needs the training data of each expert, and no extra labels.
> > >   * Each of the four embedding vectors serves as a representation for the knowledge the respective expert has learnt
> > > * During training, the Nexus router is not trained directly, but computed at each step as the output of the MLP hypernetwork, and the MLP gets trained via backpropagation, so that it learns to maps the inputs (=the stacked embeddings of the four experts’ training data) to useful representations, which tell the model which expert would be best for a given token.
> > > * At the end of the training, we can now optimize our model for inference as follows:
> > > In the router layer of each transformer block, we pass the expert dataset embeddings one more time through the trained MLP hypernetwork. **The output (=router layer), which is usually computed in every training step, can now be saved with the model weights**, as it will not change anymore (the inputs were always constant, and the MLP and all other weights will not be updated anymore during inference).
> > >
> > > Second, considering the inference setting, where Nexus and vanilla MoE behave identical:
> > > * The vanilla MoE and Nexus router forward pass is exactly the same. Specifically, they have the same number of parameters, use the same amount of FLOPs, and receive the same inputs. The difference between them is the mechanism of how their router layer was learnt during training.
> > > * Note that viewing the routing operation as **A)** a linear projection from inputs to *n* router probabilities (vanilla MoE) or **B)** computing the dot-product similarity between the inputs and each of the *n* columns of the router matrix (Nexus) are just two sides of the same coin, as mathematically and implementation-wise, these are exactly the same. Therefore, our approach also offers a novel perspective to understanding the routing operation.
> > >
> > > To summarize, while the core strength of Nexus is that it utilizes the available information about expert training data to improve the router training, at inference time it does not have any advantages and does not receive additional information or labels.
> > >
> > > > **The authors will not publish a fully usage code base to facilitate future research**
> > >
> > > We very much agree with **R sqxx** on the importance of openly available research and reproducible experiments. Making the source code for the experiments available is often a practical way to do so, but might not always be possible for various reasons. Experiments performed using non-standard training frameworks might require dependencies whose release is beyond the authors control, due to copyright or intellectual property restrictions.
> > >
> > > This is why the ICLR author guidelines [1] suggest, but don’t require authors to submit code as part of the submission. For this paper, we chose the best available options to ensure reproducibility, by:
> > >
> > > * Using an openly available dataset for our experiments (SlimPajama [2])
> > > * Describing our architecture and specifically our routing algorithm in great detail [3]
> > > * Describing the model and training setting in great detail [4]
> > >
> > > These steps ensure that our experiments are transparent and reproducible within any other training framework. If **R sqxx** has further questions on reproducing our experiments which are not answered yet in the paper, we are happy to amend and extend the relevant sections of the paper. Otherwise, we would kindly ask **R sqxx** to not let a factor outside of authors control influence their evaluation of the validity and relevance of this paper.
> > >
> > > [1] [ICLR 2025 Author Guide](https://iclr.cc/Conferences/2025/AuthorGuide)
> > >
> > > [2] https://huggingface.co/datasets/cerebras/SlimPajama-627B
> > >
> > > [3] Section 3 in the manuscript for a general description, Appendix A for the detailed routing algorithm.
> > >
> > > [4] Sections 4 in the manuscript

---

> > > > ### Author Response · Authors · 2024-11-25
> > > > **Official Comment by Authors (continuation)**
> > > >
> > > > > **The domain embedding need to be investigated more comprehensively; some suggestions are…**
> > > >
> > > > We thank **R sqxx** for highlighting this point about the robustness of our approach. We address their points separately:
> > > >
> > > > >> **...the usage of different embedding models**
> > > >
> > > > We choose a state-of-the-art embedding model, and average the embeddings of each document over the whole training data of an expert. Therefore, we expect all strong embedding models that can extract useful features from the documents to work well with our approach.
> > > > Furthermore, the MLP hypernetwork transforms the embeddings before they are used in the router, and can easily be adjusted to embeddings of any dimensionality before starting the training.
> > > >
> > > > >> **...the sensitivity to mislabeling, or the effect of different labeling techniques**
> > > >
> > > > We expect our approach to work robustly regardless of the exact assignment of labels, as long as each labeled data group relates roughly to a distinct capability/knowledge domain, as the domain labels are exclusively used to separate the SlimPajama dataset into *n* folds for the *n* experts.
> > > >
> > > > * Note that these labels can be assumed to be ground truth and free of mislabeling, as they are describing the source of the data instead of the content. For example, a single Wikipedia article containing excerpts from a famous book can not be seen as “mislabeled” even though the majority of book content is assigned to the “Books” fold of SlimPajama, because by definition, this label applies to all data from Wikipedia.
> > > > * The dataset embeddings therefore only describe a “tendency” of this dataset—and therefore of the expert trained on this dataset—to be competent on certain subjects, but do not and can not offer a “sharp” separation of topics due to the interdependencies of different domains of knowledge. This would not be enough on its own to perform token-level routing, but serves as a helpful inductive bias for the router, which leads to Nexus learning better after upcycling [1] and adapting faster to new domains [2].
> > > > * However, substantial overlap between the distribution of clusters would degrade the effectiveness of our approach, as in this case, all the experts would learn roughly the same distribution. For the SlimPajama dataset, we inspected samples and summary statistics from each fold thoroughly, to make sure that the majority of documents does not have overlapping features. This would be easy to repeat for a new dataset.
> > > >
> > > > So to summarize, as long as the labeling technique can create somewhat distinct data folds, the experts trained on those folds will learn different capabilities, and Nexus can utilize this by routing based on the representation of this distinct training data fold.
> > > > [1] Table 1/Figure 2 in the manuscript
> > > > [2] Figure 3 in the manuscript
> > > >
> > > > > **The authors suggested c-BTM, however this work seems to be under review and it is not guaranteed to provide useful domain information for SMoE training.**
> > > >
> > > > We thank **R sqxx** for raising critical questions about the clustering technique in the c-BTM paper. In our exploratory analysis, we considered using unsupervised clustering for domain discovery, but in the end decided to use the inherent domains of SlimPajama instead. We would be delighted to see future work tackle the issue of creating separate training folds in an unsupervised manner, but would like to note that **A)** the question of how to obtain the different training folds is not related to the embedding step in Nexus, which can be performed independently of how the expert training data was created, and **B)** new techniques for discovering data folds might be less relevant for practitioners, who often don’t have one big, mixed dataset, but already are in possession of many small datasets from different sources which are ideal for the expert training of Nexus.
> > > >
> > > > We thank **R sqxx** for the constructive discussion and are happy to clarify any other points. If this satisfyingly addresses the open points regarding the domain embeddings, we would kindly ask reviewer **sqxx** to consider increasing their score to reflect this.

---

> > > ### Author Response · Authors · 2024-12-01
> > >
> > > Following up on the constructive suggestions by **R sqxx**, we present **additional evidence** for the robustness of our approach. Specifically, we performed experiments with a different technique for creating the expert datasets, to test the **“sensitivity to mislabeling, or the effect of different labeling techniques”**. This showcases that the Nexus router is robustly better than the vanilla MoE baseline, strengthening our highlighted advantages of Nexus. In the following, we report the setup and results for both new experiments.
> > >
> > > ---
> > >
> > > Experiment design
> > > ====
> > >
> > > **Experiment A – "pretraining"** (following Section 4.1, points 1+2 in the manuscript):
> > >
> > > 1. Cluster the SlimPajama dataset via balanced k-means clustering into 8 roughly equal clusters
> > > 2. This uses the clusterer from the BTX [1] paper
> > > 3. Train a dense 470M expert on each of the 8 clusters for 25B tokens
> > > 4. Upcycle the experts into a vanilla MoE baseline (BTX [1]) and a Nexus model (ours)
> > > 5. Train the upcycled models each for 25B tokens on all SlimPajama data
> > >
> > > **Experiment B – "adding a new expert"** (following Section 4.1, point 3 in the manuscript):
> > >
> > > 6. Train an additional 470M expert for 5B tokens on math data, to simulate adding a new domain
> > > 7. Append the expert to the MoE layers of both the trained vanilla MoE baseline and the Nexus model **from experiment A after step 4**
> > > 8. Train the resulting models each for 5B tokens on a mix of SlimPajama and the math dataset
> > >
> > > **Differences to previous experiments:**
> > > * Instead of using the inherent domains of SlimPajama, domains are now created through unsupervised clustering
> > > * This tests **“sensitivity to mislabeling, or the effect of different labeling techniques”**, as proposed by **R sqxx**, by using a different, much noisier labeling technique (unsupervised clustering).
> > >
> > > ---
> > >
> > > Results
> > > ====
> > >
> > > **Results – A:**
> > > |       | Vanilla MoE | Nexus  |
> > > |---------------|-------------|--------|
> > > | all clusters  | 11.0      | **10.9** |
> > > | math          | 9.8       | **9.6**  |
> > >
> > > We evaluate the perplexity of both models on the test split of the whole SlimPajama dataset (“all clusters”) and a new dataset (“math”).
> > > Note that the model has not seen the math data yet at this point.
> > >
> > > **Results – B**:
> > > |       | Vanilla MoE | Nexus  |
> > > |---------------|-------------|--------|
> > > | all clusters  | 12.4      | **11.3** |
> > > | math          | 6.4       | **6.0**  |
> > >
> > > We evaluate the perplexity of both models on the test split of the whole SlimPajama dataset (“all clusters”) and the new dataset (“math”), after adding the math expert and finetuning both models.
> > >
> > > ---
> > >
> > > **Discussion**
> > > ====
> > > For pretraining (experiment/results **A**) Nexus performs better after upcycling than the vanilla MoE model: 10.901 (Nexus) vs. 10.995 (vanilla MoE), lower is better
> > > After adding a new expert (experiment/results **B**), Nexus adapts much better to the new domain, while retaining more of the performance on previous domains compared to the vanilla MoE baseline.
> > >
> > > We hope this provides convincing evidence to **R sqxx** about the robustness of our method when changing the labeling technique, and are happy to address follow-up questions in the remaining time of the rebuttal period. We thank **R sqxx** for proposing this experiment, which combined with our previous experiments reported in the manuscript provides better evidence for the effectiveness of our method. If this addressed all open points of **R sqxx**, we would appreciate it if they increase their score to reflect this accordingly.
> > >
> > >
> > > [1] Sukhbaatar, S., Golovneva, O., Sharma, V., Xu, H., Lin, X. V., Rozière, B., Kahn, J., Li, D., tau Wen-Yih, Weston, J., & Li, X. (2024). Branch-Train-MiX: Mixing Expert LLMs into a Mixture-of-Experts LLM. https://arxiv.org/abs/2403.07816

---

### Official Review · Reviewer_Ykss · 2024-11-05

**Soundness:** 2
**Presentation:** 3
**Contribution:** 2
**Rating:** 6
**Confidence:** 4

**Summary:**

This paper introduces Nexus, a method for expanding dense models into Mixture-of-Experts (MoE) models, along with a novel router that introduces strong domain inductive bias and encourages expert differentiation. The experiments validate the advantages of this router in merging separately trained dense experts into an MoE model and in extending to new domains, compared to directly merging models and using a standard linear router. The analysis experiments demonstrate that Nexus achieves domain-specialized routing and shows better performance than dense models under different load balancing and training data ratio settings.

**Strengths:**

- The paper is clearly written and easy to understand.
- The analysis experiments show that the proposed method indeed encourages experts to specialize.
- The method of incorporating domain embedding information is straightforward and has been shown to outperform linear routers and model merging through comparative experiments.

**Weaknesses:**

- The comparison to related work is inaccurate. In lines 516-519, the paper discusses ensemble methods, stating that these methods require computing all experts rather than sparse activation. However, both BTM and C-BTM compare results using different top-k values after ensembling model groups, and C-BTM even outperforms MoE with the same activation parameter count. If Nexus emphasizes the importance of specialization, these methods that conditionally ensemble specialized LLM experts should also be compared.
- The comparison to MoE baselines is not entirely reasonable. The Nexus router has more parameters than the linear router. Using the MLP-router from ModuleFormer as the baseline might be more appropriate to exclude the influence of parameter differences. Additionally, in Experiment 5.2, it might be more suitable to expand the parameters of the linear router's weights after expanding the experts rather than resetting them. The poor performance of the linear router at 200M tokens in Figure 3 (left) might be due to the randomly initialized router being undertrained.
- The paper does not explain the necessity of composing domain-specific experts into experts. It only compares merging dense models and initializing MoE with dense experts, but does not compare:
   1. Training a dense model with the mixed data described in the paper. Since the number of tokens used to train the seed model (25B/40B) is similar to the number of tokens used for subsequent domain-specific training, the seed model may not have converged, and domain-specific training can cause significant changes to the model. Merging specialized models at this point might degrade model performance. The authors could consider adding results from training a dense model on all data (including initial and subsequent data) to better illustrate the significance of converting dense models to MoE.
   2. Experiments based on a stronger pretrained foundation model, similar to Branch-Train-Mix. The ratio of training data for the seed model and expert models in the paper differs significantly from real-world domain extension scenarios. Due to the large gap between expert models and the seed model mentioned in point 1, the conclusions drawn from training after merging might differ significantly from real-world scenarios.
   3. Showing the performance of each expert model on the corresponding domain task in Table 1.
- Figures 4 and 5 lack baseline comparisons when explaining specialization. Since experts are already specialized and have clear preferences, it is reasonable to see distinct biases. Comparisons with the linear router would further support these findings.

**Questions:**

- The performance of the code domain decreases after the code expert is merged into the MoE model. Do experts from other domains have similar issues?

---

> ### Author Response · Authors · 2024-11-21
> **Author reply**
>
> We would like to thank R Ykss for their thoughtful and constructive feedback. The comments have been very helpful in improving the clarity and depth of our work.
>
> We want to address the points raised one by one:
>
> > **The paper does not explain the necessity of composing domain-specific experts into experts**
>
> We would like to address the subpoints of this issue by providing additional evidence and resolving some misconceptions:
> >> **The authors could consider adding results from training a dense model on all data (including initial and subsequent data) to better illustrate the significance of converting dense models to MoE.**
>
> We thank the reviewer for the suggestion of this baseline, which can provide additional context to the performance of the Nexus method. We ran the experiment and report the results in the global response, Table a) (also included in the new Appendix F, Table 6). The 2.8B Nexus model outperforms the continued pretraining baseline clearly, while also providing the benefits of extensibility with new experts for easy adaptation to new domains. For more details, we refer to the global response.
>
> >> **[The authors could consider] showing the performance of each expert model on the corresponding domain task**
>
> As a response to the helpful suggestion, we systematically evaluated each expert on each domain and updated our paper and include a table of expert performance by domain in Appendix E, Table 5.
>
> >>  **The number of tokens used to train the seed model (25B/40B) is similar to the number of tokens used for subsequent domain-specific training, the seed model may not have converged. [The authors could consider adding] experiments based on a stronger pretrained foundation model**
>
> Regarding the number of tokens used to train the seed model: the 470M/2.8B seed models were not, as assumed by R Ykss, trained on 25B/40B tokens, but instead both models were pretrained on 750B tokens, which at this scale is sufficient for the model to converge. Section 4.1 describes the training process. We have slightly updated it to make our training setting more easily understandable.
>
> > **Figures 4 and 5 lack baseline comparisons when explaining specialization.**
>
> We thank R Ykss for this helpful feedback, and have ran the necessary analysis. The paper has been updated with Figures 9 and 10 in Appendix H, which show baselines for Figures 4 and 5, respectively. When comparing the routing distribution of Nexus with the baseline linear router, we find that for the training domains, both models achieve good specialization with high probability to the experts own domains. However, for the new domain (Code), Nexus demonstrates significantly higher specialization (~0.7 vs ~0.3) which also correlates with higher performance in this new domain.
>
> > **The performance of the code domain decreases after the code expert is merged into the MoE model. Do experts from other domains have similar issues?**
>
> R Ykss is indeed correct in that the Nexus model with merged code expert initially performs worse than the code expert on code generations tasks. This is not surprising as the code expert is solely specialized in this domain with a high performance on the code task, but has lower performance on the other tasks (“catastrophic forgetting” as seen in the new Appendix E, Table 5). The advantage of Nexus is to adapt to new domains without regression on old domains. Furthermore, the merging procedure itself averages the non-FFN parameters of the models to merge, and we observe that this can temporarily decrease the performance on some tasks. However, after a small amount of training the model recovers.
> For the other domain experts, this effect is not visible because other domains do not directly correlate with any particular downstream tasks, except the Book expert, which slightly outperforms the final Nexus on MMLU (39.6 vs 39.4)
>
> > **The comparison to MoE baselines is not entirely reasonable. The Nexus router has more parameters than the linear router**
>
> We thank R Ykss for pointing out the mismatch in total parameters in the models. However, we would like to clarify an important point, as the MLP in the Nexus router is subtly different from using an MLP in place of the linear router projection:
> The additional MLP in Nexus acts as a hypernetwork with the role of projecting domain embeddings to expert embeddings. Therefore output of this layer (i.e. hypernetwork) is used as the weights for the linear router.
> Most importantly, during inference, the MLP parameters do not change anymore, and neither do the inputs (=dataset embeddings). This means we can precompute the constant output of the MLP, and “hardcode” this as router weights of the model. This means that the architecture and parameter count of Nexus during inference are **exactly identical** to a vanilla MoE model.
> Finally, this additional layer adds less than 1% additional parameters compared to the overall parameter count of the model (see Table b) in the global response).

---

> ### Author Response · Authors · 2024-11-21
> **Author reply (continuation)**
>
> > **The comparison to related work is inaccurate, BTM and c-BTM can be evaluated sparsely**
>
> We thank R Ykss for pointing this out. We revised the related work with a note about how BTM and c-BTM can be used sparsely.
>
> However, even though both BTM and c-BTM could be used with top-k, this comes with unique drawbacks. Firstly, each expert in BTM / c-BTM retains a separate set of attention weights for all experts, which results in substantially higher memory requirements and an increased total/active parameter count compared to Nexus (e.g.  5 times the 2.8B seed model for c-BTM = 13.7B total parameters, instead of 9.1B total parameters for Nexus with 5 experts). Particularly, having a separate KV cache for each expert would impact inference times significantly at scale. Secondly, while both BTM and c-BTM allow top-k computations, they are restricted to using the same experts ensemble for all tokens in the sequence, while vanilla MoE and Nexus offer token-wise routing. For BTM, calculating the ensemble weights even requires a forward pass on the prompt for every expert, which breaks sparsity, and for c-BTM, a separate embedding model has to be trained and evaluated during inference. To summarize, as we propose a new MoE architecture, we mainly compare our method with linear MoE architectures like BTX [1] that are directly comparable in terms of memory and compute requirements.
>
> [1] Sukhbaatar, S., Golovneva, O., Sharma, V., Xu, H., Lin, X. V., Rozière, B., Kahn, J., Li, D., tau Wen-Yih, Weston, J., & Li, X. (2024). Branch-Train-MiX: Mixing Expert LLMs into a Mixture-of-Experts LLM. https://arxiv.org/abs/2403.07816
>
> > **In Experiment 5.2, it might be more suitable to expand the parameters of the linear router's weights after expanding the experts rather than resetting them. The poor performance of the linear router at 200M tokens in Figure 3 (left) might be due to the randomly initialized router being undertrained.**
>
> We thank R Ykss for this suggestion, and will include this additional experiment in the camera-ready version of the paper.
>
> As a response to these very helpful comments, we have slightly clarified the sections in the paper touched by these issues, and added Tables 3, 5, and 6, and Figures 9 and 10 to the Appendix. We are happy to provide any further information if needed.

---

> > ### Comment · Reviewer_Ykss · 2024-11-26
> >
> > - I appreciate the authors' further explanations. I agree with the author that the performance gap between sparse model and the seed dense model will increase. Demonstrating the performance gap between Nexus and linear MoE with data scaling would further highlight the effectiveness of Nexus.
> >
> > - Additionally, regarding the experiments on expanding FFN-width, my main concern is that most upcycling works, especially when top-k is set to 2 or higher, resulting in increased FLOPs, does not consider a simple and potentially strong baseline: increasing the FLOPs of dense models through methods such as copying, to make them comparable with sparse models. If the performance of dense models could be demonstrated in a computation-comparable setting (for example, expanding the dense model and training it on half of the data used for expert training, and then conducting the same training), especially if it is found that this baseline performs comparably to the linear router MoE but weaker than Nexus, this would more strongly support the effectiveness of Nexus.

---

### Author Response · Authors · 2024-11-21
**General response to reviewers**

We greatly appreciate the thoughtful and detailed feedback from the reviewers. To the best of our knowledge, our work is first to present a Mixture of Experts architecture that ticks all boxes of (1) efficient, fully parallel training with no gradient synchronization between experts (2) specialized experts with human-interpretable domains (3) adaptability to new domains without forgetting of old domains. Regarding these goals of efficiency, specialization, and adaptability, the reviewers commented that “the paper provides a novel router based on expert embeddings that outperform previous approaches for combining dense experts” [fKKY], “The analysis experiments demonstrate that Nexus achieves domain-specialized routing” [Ykss], and ”The results show that Nexus can offer some performance gains while improving the specialization and adaptability” [sqxx].

We are further encouraged that reviewers found the question is critical and important for the field: “SMoE is an important and interesting research direction. This work shows that improving experts specialization can be helpful and can generalize the new domains” [sqxx].
Regarding our method, reviewer Ykss found that “The method of incorporating domain embedding information is straightforward and has been shown to outperform linear routers and model merging through comparative experiments.”, while reviewer sqxx highlighted that “the method is simple and intuitive sound. The results are encouraging and can corroborate the motivation.”, also supported by us “training LLMs from scratch at a reasonable scale” [sqxx].

We also thank the reviewers for their constructive critical feedback. We have provided detailed explanations addressing each of their points including additional evaluations that were requested, demonstrating how our work robustly improves over existing architectures.

Firstly, we present a series of new baselines, that compare Nexus to continued pretraining of dense models in a data-matched setting, as suggested by reviewers Ykss and sqxx. This means that the seed models (which initially both were pretrained for 750B tokens) are further trained on the combined data of all individual experts and additionally on the same amount of finetuning data that we use to finetune Nexus after upcycling from the experts.
|                              | Know. | Science | Reason. | MMLU | Avg.  |
|------------------------------|-------|---------|---------|------|-------|
| **470M Models**              |       |         |         |      |       |
| Seed Model                   | 14.0  | 51.4    | 50.5    | 29.8 | 36.4  |
| Seed Model + 145B tokens     | **19.9**  | 53.8    | 50.8    | 29.6 | **38.5**  |
| Nexus                        | 16.7  | **55.0**    | **52.3**    | **29.8** | **38.5**  |
|                              |       |         |         |      |       |
| **2.8B Models**              |       |         |         |      |       |
| Seed Model                   | 27.1  | 62.0    | **63.8**    | 35.4 | 47.1  |
| Seed Model + 200B tokens     | 28.8  | 66.4    | 62.7    | 41.4 | 49.8  |
| Nexus                        | **33.2**  | **67.3**    | 62.6    | **39.4** | **50.6**  |

Table a): The Nexus model based on the 470M parameter experts outperforms the continued training baseline with the 470M seed model on 3 out of 4 categories and scores equal in the overall average. The Nexus model based on the 2.8B parameter experts outperforms the respective baseline in 3 out of 4 categories and clearly outperforms in the overall average. Besides that, the Nexus model can be trained in less wallclock time as all experts can be efficiently trained in parallel without synchronizing gradients, and Nexus is adaptable to new domains without losing performance on old domains, whereas dense models often run into the “catastrophic forgetting” problem when training on new data.

---

> ### Author Response · Authors · 2024-11-21
>
> Furthermore, we present an overview of total and active parameters, from which memory need and FLOPs can be inferred, as requested by reviewer sqxx.
> |                          | Total Parameters | Active Parameters (Training) | Active Parameters (Inference) |
> |--------------------------|------------------|------------------------------|-------------------------------|
> | **470M Models**          |                  |                              |                               |
> | Seed Model (470M)        | 467,682,304      | 467,682,304                  | 467,682,304                   |
> | MoE (Linear routing)     | 1,298,252,800    | 606,110,720                  | 606,110,720                   |
> | Nexus                    | 1,312,834,560    | 620,692,480                  | 606,110,720                   |
> |                          |                  |                              |                               |
> | **2.8B Models**          |                  |                              |                               |
> | Seed Model               | 2,752,565,760    | 2,752,565,760                | 2,752,565,760                 |
> | MoE (Linear routing)     | 9,044,226,560    | 4,325,429,760                | 4,325,429,760                 |
> | Nexus                    | 9,129,218,560    | 4,410,421,760                | 4,325,429,760                 |
>
> Table b): During training, Nexus has less than 1% additional parameters compared to the vanilla MoE, leading to no measurable increase in step times. During inference, they are even exactly equal, as the output of passing the constant dataset embeddings (1 per expert) through the frozen MLP hypernetwork can be precomputed and hardcoded as the router weights. Nevertheless, Nexus keeps the advantage of being able to flexibly recompute the router weights if a new expert is added.
>
>
> In the individual responses to reviewers, we provide additional new data and clarification.
>
> We also update the following parts of the paper:
> * New Table 3 in Appendix B, showcasing the parameter counts of Nexus and baselines
> * New Table 5 in Appendix E, showcasing the individual performance of each expert on each domain
> * New Table 6 in Appendix F, showcasing the performance of continual training of the seed models
> * New Figures 9 and 10 in Appendix F, showcasing the routing distribution of the linear MoE router, as compared to the existing Figures 4 and 5 for Nexus.
> * Minor changes to clarify some sections
>
> We look forward to engaging in meaningful discussion and welcome any additional questions at any point in time.

---

> > ### Comment · Reviewer_Ykss · 2024-11-22
> >
> > - Thanks for addressing my concerns regarding the router parameters and training sufficiency. I find the author's discussion on the scalability inference issues brought by the separate KV cache in BTM series models very valuable, as it highlights new challenges that may arise when using model group/forest methods in the era of LLMs. Based on these contents, I have increased my score.
> > - However, in the additional baselines provided, it can be seen that the seed model, after continuing training with the full amount of tokens, performs almost on par with the linear MoE router baseline, while using significantly fewer active parameters. If the author could further provide baselines with expanded parameter counts (for example, doubling the width of the FFN blocks, or using upcycling to expand to 2 experts with top-k = 2), this would better highlight the advantages of Nexus and also demonstrate the issues with direct upcycling.

---

> > > ### Author Response · Authors · 2024-11-22
> > > **Reply to Comment by Reviewer Ykss**
> > >
> > > We thank R Ykss for contributing to the quality of this work with their insightful questions and suggestions, and appreciate the increased score.
> > >
> > > Regarding the suggested additional baseline, while we acknowledge that the close performance of the continued pretraining and linear MoE baselines is worth noting, we think this might be an artifact of our peculiar setting. To highlight the flexibility and adaptability of Nexus in the initial upcycling setting as well as when adding a new domain, we use the majority of compute in the expert training phase. However, if all models were finetuned for longer, we expect the gap between both Nexus and the linear MoE baseline on the one hand, and the continued pretraining baseline on the other hand, to increase, as the sparse models are more sample-efficient. Furthermore note that continued pretraining of the seed model with doubled width of the FFN layer would still be data matched, but not compute matched anymore to the expert training stage, as in this case the seed model would have more active parameters than the individual experts.

---

### Meta-Review · Area_Chair_Npvy · 2024-12-20

**Metareview:**

This paper introduces Nexus, a method for expanding dense models into Mixture-of-Experts (MoE) models, along with a novel router that introduces strong domain inductive bias and encourages expert differentiation.

Although the domain inductive bias can help to improve performance of the model, the paper suffers from the following shortcomings:
1.  The technical contribution of this work is rather shallow, i.e., it is not very clear to researchers to adopt Nexus to their work due to many uncertainties regarding the domain embedding designs. [Reviewer fKKY, Reviewer sqxx]
2. To one of the questions by Ykss,  authors respond that if we //However, if all models were finetuned for longer, we expect the gap between both Nexus and the linear MoE baseline on the one hand, and the continued pretraining baseline on the other hand, to increase, as the sparse models are more sample-efficient. Furthermore note that continued pretraining of the seed model with doubled width of the FFN layer would still be data matched, but not compute matched anymore to the expert training stage, as in this case the seed model would have more active parameters than the individual experts.//. It may not be appropriate to make conclusions based on such speculations, unless demonstrated empirically or theoretically.

Essentially, this paper introduces a good idea of using domain specific embeddings and allows for continual learning through allowing addition of new experts after initial training. However, this paper might need a lot of baseline experiments and more studiy/analysis before being published. Hope the reviewer's feedback can help the authors to revise their manuscript.

**Additional Comments On Reviewer Discussion:**

Reviewers agree that while some of their comments have been addressed, they see some flaws with the paper and method that have not been adequately addressed by the authors.

---

### Decision · Program_Chairs · 2025-01-22

Reject